# Different HLA Alleles Frequencies and Their Association with Clinical Phenotypes of Acute Respiratory Infections in Children

**DOI:** 10.3390/v17111495

**Published:** 2025-11-12

**Authors:** Natalia V. Palyanova, Olesia V. Ohlopkova, Alexey D. Moshkin, Kristina A. Stolbunova, Marina A. Stepanyuk, Ivan A. Sobolev, Olga G. Kurskaya, Alexander M. Shestopalov

**Affiliations:** Federal State Budgetary Scientific Institution “Federal Research Center for Fundamental and Translational Medicine”, 630117 Novosibirsk, Russia; ohlopkova.lesia@yandex.ru (O.V.O.);

**Keywords:** HLA, SARS-CoV-2, influenza, ARI, children, NGS sequencing

## Abstract

The histocompatibility gene complex plays a vital role in the body’s immune response to infections. In this work, we analyzed clinical data for 195 children hospitalized with signs of ARI in Siberia and performed genetic analysis for them. Genotyping was performed by high-throughput sequencing (NGS) using the HLA-Expert kit on the MiSeq Illumina platform. The frequencies of HLA allelic variants were calculated for each variant. For the variants detected in 20 patients or more, odds ratios (OR) were calculated for two pairs of conditions: severe/non-severe course of ARI and hypoxia/no hypoxia on admission. Six allelic variants were identified for which the odds ratio showed a significant (*p* < 0.05) association with one of the conditions. Allele HLA-A*11:01:01G is associated (OR = 5.654, 95% CI 1.631–19.600) with severe ARVI, which is consistent with the literature data, and HLA-A*03:01:01G allele is associated with ARVI without hypoxia in children (OR = 0.317, 95% CI 0.110–0.914). Alleles HLA-B*51:01:01G (OR = 4.457, 95% CI 1.355–14.663) and HLA-C*01:02:01G (OR = 4.743, 95% CI 1.538–14.629) are associated with severe ARI. HLA-DPB1*04:02:01G (OR = 0.462, 95% CI 0.244–0.876) is associated with ARI without hypoxia and HLA-DQA1*01:02:01G (OR = 1.811, 95% CI 1.003–3.268) is associated with ARI with hypoxia.

## 1. Introduction

Infectious diseases are one of the main causes of death and morbidity among people worldwide [1]. A special place in the structure of morbidity is occupied by acute respiratory infections, with a significant proportion of those affected being children [2]. Most cases of respiratory illness are mild, but some children may require hospitalization [2]. The search for the causes of severe acute respiratory infections is a current urgent problem, especially taking into account the emergence of SARS-CoV-2 and the ongoing annual influenza epidemics [3]. The SARS-CoV-2 pandemic has led to a large number of studies aimed at finding an answer to the question of why some people easily tolerate COVID-19, while others die. However, this question should be expanded to other respiratory infections.

Possible causes of different courses of respiratory diseases may include the patient’s genetic characteristics, which determine specific responses of the immune system to infection, since infectious diseases are the main selective pressure on genes involved in the immune response [4,5,6]. The genes themselves, which implement a diverse immunological response to a wide range of infectious pathogens, are the most numerous and most variable in the human genome and are called HLA (human leukocyte antigens) or MHC (major histocompatibility complex) for other species [6]. HLA genotyping and identification of patients with alleles associated with severe respiratory diseases or with a risk of hypoxic conditions will help the doctor decide on treatment tactics and prescribe treatment for such patients without waiting for the condition to worsen. Genetic work is quite complex, labor-intensive, and expensive, and the results may vary depending on the population. Our work is pilot research to study the relationship between patient genotypes and the severity of respiratory diseases in Siberia.

The genes encoding components for HLA molecules are located on chromosome 6 and have many variants, with more than 9000 alleles distributed in the population with varying frequencies [7]. MHC molecules are responsible for presenting antigens, including those from viruses to T cells, triggering an immune response [3]. HLA class I molecules (HLA-A, HLA-B, and HLA-C) are present on the surface of all cell types, except red blood cells and trophoblast cells, and present peptides from the cytoplasm to the cell surface (including viral or bacterial peptides during infection). HLA class II molecules (HLA-DR, HLA-DQ, and HLA-DP) are found on the surface of antigen-presenting cells (dendritic cells, macrophages, B lymphocytes). The most convincing associations between classical HLA class I and class II alleles and infectious diseases have been reported for chronic viral infections [8]. For example, high heterozygosity is associated with slower progression of AIDS [9] and a higher likelihood of overcoming hepatitis B [10].

Long-term studies have revealed certain patterns and alleles that determine mild or severe courses of acute respiratory viral diseases. The protective role of the HLA genotypes HLA-DRB1*01*04, HLA-DRB1*07*08, and HLA-DRB1*11*13 against influenza AH1N1pdm2009 has been described in the literature [11]. HLA-A*11, HLA-B*35, and HLA-DRB1*10 are associated with a predisposition to a severe course of H1N1 influenza [3,12]. The presence of HLA-DRB1*07*15 in the genotype of pregnant women is associated with a severe course of influenza [11]. The combined presence of genotypes HLA-F7:0976GA, HLA-PAI-1:6754G4G and HLA-DRB1*07*15, HLA-DRB1*16*16, HLA-DRB1*01*01 may be a predictor of the development of severe forms of influenza [11]. The severity of SARS-CoV-1 infection is significantly associated with HLA-B*46:01 [13]. HLA-DQB1*03:02 is significantly associated with moderate or mild MERS-CoV disease [14]. Also, recently, many studies have appeared revealing a link between mild/severe SARS-CoV-2 and the patient’s HLA genes. Certain HLA gene variants may cause the formation of a “cytokine storm” in patients with COVID-19 [3]. HLA-B*15:03 demonstrated the greatest ability to present SARS-CoV-2 peptides that are common to human coronaviruses, suggesting its potential to mediate cross-protective T cell immunity. Other alleles with high predicted binding capacity include HLA-A*02:02 and HLA-C*12:03 [15]. The HLA-B*15:01 allele is also associated with asymptomatic SARS-CoV-2 [16], and the HLA-A*11:01:01:01 and HLA-DRB1*09:01 alleles are associated with severe COVID-19 [17,18]. Also, the HLA-A*01:01 allele was associated with high risk of severe COVID-19, while patients with HLA-A*02:01 and HLA-A*03:01 had low risk [19].

The aim of this study was to determine the allelic composition and allele frequencies in children hospitalized with signs of acute respiratory infection, as well as to identify the relationship between the disease phenotype and the patient’s genotype.

## 2. Materials and Methods

### 2.1. Clinical Data Collection and Patients Selection

Clinical data were obtained from patients aged 0 to 16 years, hospitalized in medical institutions of Novosibirsk from October 2023 to June 2024 with signs of acute respiratory viral infection, a total of 107 boys and 88 girls. Voluntary informed consent was obtained from legal representatives for all study participants. The study was approved by the Biomedical Ethics Committee at the Federal Research Center for Fundamental and Translational Medicine. All patients (or their legal representatives) filled out a questionnaire upon admission to the medical institution, containing questions about the patient’s age, gender, condition, and concomitant diseases. The study involved 27 adolescents aged 10–16 years and 168 children aged 9 years and younger. All patients examined denied smoking. Chronic diseases were reported by 13 patients. Inclusion criteria were age under 16 years, hospitalization with symptoms of acute respiratory infection, and preliminary diagnosis of viral infection based on anamnesis and blood test. We excluded from the study 7 patients with identified concomitant HIV infection, mycoplasma infection, and congenital respiratory diseases.

During the initial examination in the emergency department, blood tests were taken, anamneses were collected, and preliminary diagnoses were made. A complete blood count was used as an additional source of information about the viral or bacterial nature of the disease, which helped to reduce the unnecessary use of antibiotics. The blood oxygen saturation was also measured, and hypoxia was assumed in patients with cyanosis, low SpO2 (under 95%), stridor breathing, and dyspnea.

The severity of the disease was determined based on clinical guidelines, which included clinical and laboratory parameters which differentiated depending on the patient’s age and included respiratory distress, obstructive airway conditions, respiratory failure, oxygen saturation < 95%, cardiovascular complications, central nervous system complications, dehydration, and resistant fever. Although hypoxia is one of the criteria for disease severity, patients in severe condition do not always suffer from hypoxia, and not all patients with hypoxia are classified as severe.

### 2.2. RT-PCR Screening Procedure

The primary diagnosis made upon admission was clarified taking into account PCR analysis for the presence of DNA/RNA of viruses that cause acute respiratory viral infections: hRSv (human respiratory Syncytial virus), hMpv (human metapneumovirus), hPiv (human parainfluenza virus-1–4), hCov (human coronavirus: OC43, E229, NL63, HKUI, SARS-CoV-2), hAdv (human adenovirus B, C and E), hRv (human rhinovirus), hBov (human bocavirus), enterovirus, influenza A, and influenza B. Nasopharyngeal swabs were taken no later than 7 days after symptom onset. The RT-PCR test was performed according to the manufacturer’s instructions (AmpliSens ARVI-screen-FL, Moscow, Russian Federation). In case of a positive PCR test, a more precise diagnosis was made indicating the virus; in case of a negative PCR test, the diagnosis was made based on the anamnesis, and laboratory and instrumental tests.

### 2.3. HLA Genotyping

The patients were randomly selected for the major histocompatibility complex—HLA genotyping among all patients with a primary diagnosis not related to bacterial infection (presumed to have viral infection). The collection of genetic material for HLA typing occurred simultaneously upon admission to the health care facility. Ethnicity was indicated by 58 patients: 48 patients were Russian, 1 was Azerbaijani, 2 were Armenian, 1 was Kazakh, 3 were Kyrgyz, 1 was Tajik, 1 was Tatar, 1 was Traveler, and the rest did not indicate their nationality.

Genetic material from patients was obtained according to generally accepted pro-tocols from buccal and pharyngeal swabs. HLA typing was performed using a reagent kit for preparing libraries of DNA fragments of HLA I (HLA-A, HLA-B, HLA-C) and II (HLA-DRB1, HLA-DPB1, HLA-DQB1, HLA-DRB3/4/5) class genes for genotyping by high-throughput sequencing (NGS) HLA-Expert (DNA technology, Moscow, Russian Federation) on the MiSeq Illumina platform (Illumina Inc., San Diego, CA, USA) according to the manufacturer’s instructions. Bioinformatics analysis was performed using specialized software HLA-Expert 2.0.

### 2.4. HLA Allele Frequencies

As part of the study, the frequencies for each allelic variant of the genes of HLA were determined: HLA-A, HLA-B, HLA-C, HLA-DRB1, HLA-DPB1, HLA-DQA1, HLA-DQB1, and HLA-DRB3/4/5. Allele frequencies were estimated by dividing the number of patients with a given allele by the total number of individuals in the group (i.e., identical alleles in homozygous patients were not taken into account) and multiplying the result by 100 to obtain frequencies expressed as percentages. We used allele frequencies from [20] as a reference sample and population control for estimation of HLA allele frequencies in the Russian population.

### 2.5. Determining the Relationship Between a Patient’s HLA Genotype and the Course of a Respiratory Disease

After calculating the frequencies for each identified allelic variant, the most common alleles were selected, among which variants associated with the development of hypoxia or with a severe/mild course of infection were identified. Since the group of adolescents was small, we established the following groups of patients:All the patients of the study, 0–16 years old, PCR both +/− (children and adolescents with ARI)Children 9 years old and younger, PCR both +/− (children with ARI)Children 9 years old and younger, PCR + (children with ARVI)

We did not select the group of adolescents separately, since there were few of them and it was not possible to obtain statistical reliability from such a number. Each group was divided into two pairs of groups: groups with mild or severe disease and groups with or without hypoxia. For each allele, the odds ratio was calculated for each pair of groups.

### 2.6. Statistical Analysis

The analysis was performed using odds ratio (OR) using four-field tables. Statistical analysis was performed using the Medstatistic online calculator [21]. Differences were considered significant at *p* < 0.05, provided that the 95% CI values did not cross 1. An OR value in the range of 0–1 corresponds to a decrease in risk, an OR greater than 1 corresponds to an increase in risk, and an OR equal to 1 corresponds to no effect.

## 3. Results

### 3.1. Clinical Data Analysis

The average duration of hospitalization was 7 days, the maximum was 16 days, and the minimum was 1 day. Respiratory diseases can be caused by both viruses and other infectious agents, so in order to choose adequate treatment it is necessary to establish the etiology of the disease. The initial diagnosis was based on the clinical picture and tests taken upon admission to the health care facility, and clarification was carried out using PCR. As a result of PCR testing for the 12 most common respiratory viruses, out of 195 patients, 132 were found to have a viral infection confirmed by PCR that causes acute respiratory viral infections (ARVI), and 47 had a negative PCR test. A total of 114 people were found to be infected with one virus; 18 people were infected with two types of viruses.

A total of 18 different diagnoses were made at discharge for this group, as listed in Table 1.

The most frequently detected were respiratory syncytial virus, metapneumovirus, coronaviruses, and bocavirus (Table 2). The most frequently co-infected viruses included respiratory syncytial virus (14), bocavirus (17), rhinovirus (14), and coronaviruses including SARS-CoV-2 (16). Co-infection was detected in 18 patients; most often it was a combination of coronaviruses or metapneumoviruses with rhinoviruses, RS virus, or adenovirus.

The clinical picture of acute respiratory disease in almost all cases was accompanied by fever (188), cough (185), malaise (115), and hypoxia (106). No cases of hemorrhagic symptoms were identified, and there were no cases of patient placement in the intensive care unit in this sample of patients. The lowest saturation of 92% was in four patients: with RS-virus, with bocavirus, with coronavirus and asthma, and one with a negative PCR. In 38 patients, saturation was less than 95%, of which 29 were diagnosed with a viral infection. In 80 patients, the saturation was 96–97%, in the remaining 77 patients, 98–99%. In our work, the most frequently detected viruses with low saturation were coronaviruses, metapneumovirus, RS virus, bocavirus, and rhinovirus. Dyspnea was present in 145 cases and was not always accompanied by hypoxia. Sore throat was present in nine patients and was associated with coronavirus, bocavirus, and respiratory syncytial virus, and PCR was negative in five cases. Headache was present in 12 cases and was observed with respiratory syncytial virus, influenza A (H3N2), bocavirus, and adenovirus; eight cases were PCR negative. Intestinal symptoms occurred in five patients and were observed with adenovirus, respiratory syncytial virus and PCR-negative patients.

### 3.2. HLA Alleles Diversity and Frequencies

In our work, we studied the diversity and frequencies of different HLA alleles in 195 children hospitalized in medical institutions of Novosibirsk with signs of acute respiratory infection and a presumptive diagnosis not associated with a bacterial infection. We identified carriers of 26 variants of gene HLA-A, 48 variants of gene HLA-B, 24 variants of gene HLA-C, 22 variants of gene HLA-DPB1, 7 variants of gene HLA-DQA1, 17 variants of gene HLA-DQB1, 35 variants of gene HLA-DRB1, 4 variants of gene HLA-DRB3, 3 variants of gene HLA-DRB4, and 4 variants of gene HLA-DRB5. For each genovariant, frequencies were calculated for all the patients. The HLA class I alleles identified in this work are presented in Table 3. In the comments column of Table 3 and Table 4, we have noted both associations identified in our work (bold font) and associations known from the literature (source references in square brackets).

Among all the identified alleles, we found variants for which associations with various courses of infectious diseases are known from the literature. Thus, variants associated with a severe course of influenza were found: HLA-A*11, HLA-B*35, HLA-DRB1*10, and HLA-DRB1*01*01 [12]. Also found were variants HLA-A*25:01 (frequency 9.36%), HLA-C*01:02, and HLA-DRB1*09:01 corresponding to severe course of SARS-CoV-2, and the HLA-DRB1*12:02 allele associated with severe course of SARS-CoV-1 [12,13]. Variants HLA-A*02:02, corresponding to a mild course of SARS-CoV-2 [19], and variants HLA-A*02:07, corresponding to a severe relapse of dengue fever [22], were not detected. Two alleles corresponding to a mild course of SARS-CoV-2 (HLA-B*15:01 and HLA-C*12:03) and one allele HLA-DQB1*03:02 associated with a mild course of MERS-CoV were also found [14]. Among our patients, 15 people were found with the HLA-B*15:01 allele, which is associated with asymptomatic SARS-CoV-2 [16]. None of them had COVID-19, but they did have bocavirus, enterovirus, rhinovirus, and RS virus.

The HLA class II alleles identified in this work are presented in Table 4.

### 3.3. Search for HLA Allelic Variants Associated with Different Clinical Phenotypes of ARI

After determining the frequencies for each identified allele, we selected the most common alleles and compared the odds ratio (OR) for conditions such as moderate/severe acute respiratory infection and whether or not there was hypoxia upon admission to the health care facility. We first checked the odds ratios for all tested patients (group ‘children and adolescents with ARI’) and identified six alleles associated with the course of acute respiratory infection. We then separated two groups: children under 10 years of age and a group of children under 10 years of age with confirmed viral infection and re-estimated the odds ratio. Table 5 shows the ORs and CIs for these alleles for all groups, statistical reliability *p* < 0.05. It was found that the A*11:01:01G allele is associated with severe acute respiratory viral infections, which is consistent with the literature data [17]. We also found that the A*03:01:01G is associated with the course of ARVI without hypoxia in children. Severe courses of ARI were associated with the B*51:01:01G and C*01:02:01G alleles in the group of children and adolescents with ARI. We also found that the A*03:01:01G and DPB1*04:02:01G alleles are associated with the course of ARI without hypoxia, and DQA1*01:02:01G with hypoxia in the group of children and adolescents with ARI.

## 4. Discussion

The clinical course of ARI in children is almost always accompanied by fever, cough, and nasal congestion, while hypoxia and shortness of breath are much less common and may require hospitalization. This study included only children hospitalized for acute respiratory infections, which indicates a moderate or severe course of the disease. Symptoms of ARVI usually do not depend on the type of virus that caused the disease, as can be seen from Table 2, because cell damage occurs in a similar manner. Respiratory syncytial virus, rhinovirus, bocavirus, and metapneumovirus remain among the most common causes of hospitalization with acute respiratory viral infections outside the influenza season. Unlike influenza and SARS-CoV-2, there is no vaccine against them, so the patient’s innate immune characteristics play a key role in determining whether the disease will progress mildly or severely. The immune response to respiratory infection also involves both universal and specific mechanisms. We suggest that a genetic predisposition to a mild or, on the contrary, severe course of the disease may be associated not only with the immune response to a specific virus (or group of viruses), but also to respiratory infections in general.

Children with mild cases were not included in this sample, but we plan to expand further work in this direction to identify genetic features associated with mild or asymptomatic cases of ARI. In this study, we did not recruit a group of “healthy volunteers”, since people suffer from respiratory diseases from childhood and it is almost impossible to find someone who has never suffered from them. The diversity of respiratory viruses also raises the question of the precise diagnosis of “viral infection”, since mutations can allow viruses not only to evade immunity, but also test systems. In our work, we tried to select patients so that the diagnosis of “viral infection” was most likely; however, to be completely clear, we write “respiratory infection” in cases where PCR is negative. Some patients tested negative in the PCR test for respiratory viruses.

In this paper, we examine genetic variants of six MHC molecules responsible for human immune responses: HLA-A, HLA-B, HLA-C, HLA-DQ, HLA-DR, and HLA-DP. One approach to identify potential genetic variants associated with resistance to infection or less severe clinical phenotypes is computer modeling, but the results obtained in this way still need to be validated against clinical data. Thus, in the work [23], the ability of different alleles to present SARS-CoV-2 antigens was modeled and it was shown that people with the HLA-A*11:01 or HLA-A*24:02 genotypes can generate effective antiviral responses mediated by T cells compared to HLA-A*02:01. On the other hand, in the work [17], the allele A*11:01:01:01 is associated with severe COVID-19. In our work, we confirm the association of severe COVID-19 and ARVI with this allele.

In this work, the ethnicity of the patients was not taken into account, since only half of the patients indicated it. The patients in our study were from Russia and belonged to several ethnic groups, mainly Russian. HLA allele and haplotype frequencies in Russia exhibit significant diversity [20,24]. It is known from the literature that the following HLA allele frequencies are characteristic of the Russian population: A*02:01:01:01 (27,1%), A*03:01:01:01 (13,0%), C*07:02:01:03 (13,1%), B*07:02:01:01 (13, 0%), A*01:01:01:01 (11,6%) и C*07:01:01:01/16 (10,4%) [20]. Among them, we identified allele A*03:01:01 and show that it is associated with the course of ARVI without hypoxia. The other common alleles from this list show no associations with the course of respiratory diseases. The most common (5%) haplotypes among representatives of the Caucasian race are HLA-A1, -B8, and -DR17 [25]. In our work, these alleles also turned out to be among the most common.

When searching for articles related to alleles that we identified as associated with severe/non-severe ARI or absence of hypoxia in ARI, only a few articles related to specific alleles were found. This is because samples of patients with infectious diseases are usually not large enough to obtain reliable results. The exception is the situation with influenza, for which statistics and data have accumulated over many years, and SARS-CoV-2 [6,7,9,12,16,18,19,20,22,26]. The vast amount of genetic data collected during the COVID-19 pandemic has allowed some alleles to be described as associated with severe or asymptomatic disease. In our work, such an allele turned out to be A*11:01:01G. In addition, we were able to find in the literature that the B*51:01:01G allele is most often mentioned in association with systemic vasculitis—Behcet’s disease [27], and the HLA-C*01:02 allele is associated with a higher risk of severe allergies, including to cephalosporins [28].

We believe that this work makes a significant contribution to understanding how genetically determined features of the immune response can influence the course of the infectious process.

## Figures and Tables

**Table 1 viruses-17-01495-t001:** The main and concomitant diagnoses of patients.

Diagnoses	Decoding Diagnoses	Number of Patients	%
	**Main diagnoses**		
J20.8	Acute bronchitis due to other specified agents	47	24.1
J20.9	Acute bronchitis, unspecified	32	16.4
J06.9	Acute upper respiratory tract infection, unspecified	23	11.8
J12.1	Respiratory syncytial virus pneumonia	17	8.7
J20.5	Acute bronchitis due to respiratory syncytial virus	17	8.7
J18.9	Pneumonia, unspecified	13	6.7
U07.1	COVID-19, virus identified	7	3.6
J10.1	Influenza with other respiratory manifestations due to seasonal influenza virus	8	4.1
J06.8	Other acute upper respiratory tract infection of multiple sites	7	3.6
J12.8	Other viral pneumonia	7	3.6
J05.0	Acute obstructive laryngitis [croup]	6	3.1
J20.6	Acute bronchitis due to rhinovirus	5	2.6
J12.3	Human metapneumovirus pneumonia	4	2.1
J12.9	Viral pneumonia, unspecified	4	2.1
J09	Influenza due to identified zoonotic or pandemic influenza virus	2	1.0
J12.0	Adenoviral pneumonia	2	1.0
J20.4	Acute bronchitis due to parainfluenza virus	1	0.5
J21.0	Acute bronchiolitis due to respiratory syncytial virus	1	0.5
	**Concomitant diagnoses**		
J45.0	Asthma, predominantly allergic	5	2.6
H66.9	Otitis media, unspecified	10	5.1
H10.9	Unspecified conjunctivitis	1	0.5
G40.9	Unspecified epilepsy	1	0.5
K29.9	Unspecified gastroduodenitis	1	0.5
K59.9	Unspecified functional disorder of intestine	1	0.5
K83.9	Unspecified diseases of biliary tract	1	0.5
L80	Vitiligo	1	0.5
K83.8	Other specified diseases of biliary tract	2	1.0

**Table 2 viruses-17-01495-t002:** Viral infections detected by PCR in patients of medical institutions in Novosibirsk.

Virus ^1^ Detected	Number of Patients	Fever	Cough	Malaise	Headache	Otitis	Sore Throat	Intestinal Symptoms	Dyspnea	Hypoxia	Nasal Congestion
hRSv	40	38	40	27	1	2	2	2	25	30	8
hMpv	22	21	22	14	0	2	0	0	11	13	4
hCov	15	15	14	11	0	1	1	0	8	10	2
hBov	15	15	15	6	1	1	1	0	5	8	5
hAdv	12	11	12	9	1	1	0	1	8	6	3
H3N2	11	11	8	9	1	1	0	0	6	5	3
hRv	11	10	11	4	0	0	0	0	5	4	8
SARS-CoV-2	7	7	7	4	0	1	0	0	3	5	1
Influenza A	6	6	6	6	0	1	0	0	2	4	1
hPiv	6	6	6	1	0	0	0	0	4	3	3
Influenza B	3	3	3	1	0	0	0	0	2	2	1
Enterovirus	2	2	2	0	0	0	0	0	0	0	2

^1^ hRSv—human respiratory syncytial virus, hMpv—human metapneumovirus, hPiv—human parainfluenza virus-1–4, hCov—human coronavirus: OC43, E229, NL63, HKUI, hAdv—human adenovirus B, C and E, hRv—human rhinovirus, hBov—human bocavirus, H3N2—influenza A/H3N2.

**Table 3 viruses-17-01495-t003:** Distribution of HLA-A*, HLA-B*, HLA-C* alleles frequencies identified in patients with ARI symptoms (number and percentage of patients with the allele, number of homozygotes).

Alleles	Number of Patients	Frequencies, %	Homozygotes	Comments
A*02:01:01G	86	42.36	13	
A*01:01:01G	49	24.14	4	
A*24:02:01G	49	24.14	1	
A*03:01:01G	46	22.66	2	**We have identified the course of ARVI without hypoxia ^1^**
A*11:01:01G	24	11.82	1	Severe course of COVID-19 [17,18], Severe course of influenza H1N1 [12] **We have identified a severe course of ARVI ^1^**
A*26:01:01G	21	10.34	0	
A*25:01:01G	19	9.36	1	
A*32:01:01G	17	8.37	0	
A*31:01:02G	16	7.88	0	
A*33:03:01G	11	5.42	0	
A*68:01:02G	10	4.93	0	
A*23:01:01G	9	4.43	0	
A*30:01:01G	6	2.96	0	
A*29:01:01G	5	2.46	0	
A*03:02:01G	5	2.46	0	
A*02:05:01G	3	1.48	0	
A*29:02:01G	3	1.48	0	
A*68:02:01G	3	1.48	0	
A*24:03:01G	1	0.49	0	
A*33:01:01G	1	0.49	0	
A*66:01:01G	1	0.49	0	
A*68:01:01G	1	0.49	0	
A*02:17:01G	1	0.49	0	
A*01:02:01G	1	0.49	0	
A*30:04:01G	1	0.49	0	
A*02:06:01G	1	0.49	0	
B*07:02:01G	37	18.23	5	
B*18:01:01G	29	14.29	1	
B*35:01:01G	26	12.81	2	Severe course of influenza H1N1 [3,12]
B*08:01:01G	21	10.34	3	
B*27:05:02G	24	11.82	0	
B*44:02:01G	22	10.84	1	
B*13:02:01G	21	10.34	2	
B*51:01:01G	20	9.85	1	Asymptomatic course of SARS-CoV-2 [16] **We have identified a severe course of ARI ^1^**
B*38:01:01G	19	9.36	0	
B*57:01:01G	14	6.90	0	
B*15:01:01G	13	6.40	0	
B*40:01:01G	11	5.42	1	
B*52:01:01G	11	5.42	0	
B*35:03:01G	10	4.93	1	
B*58:01:01G	10	4.93	0	
B*41:02:01G	10	4.93	0	
B*27:02:01G	7	3.45	0	
B*27:02:01G	7	3.45	0	
B*40:02:01G	7	3.45	0	
B*55:01:01G	7	3.45	0	
B*39:01:01G	6	2.96	1	
B*44:03:01G	6	2.96	0	
B*49:01:01G	6	2.96	0	
B*56:01:01G	5	2.46	0	
B*50:01:01G	5	2.46	0	
B*15:17:01G	4	1.97	0	
B*07:05:01G	4	1.97	0	
B*37:01:01G	4	1.97	0	
B*14:02:01G	3	1.48	0	
B*40:06:01G	3	1.48	0	
B*48:01:01G	2	0.99	0	
B*13:01:01G	2	0.99	0	
B*35:02:01G	2	0.99	0	
B*39:06:02G	2	0.99	0	
B*44:03:02G	2	0.99	0	
B*53:01:01G	2	0.99	0	
B*45:01:01G	1	0.49	0	
B*15:16:01G	1	0.49	0	
B*35:08:01G	1	0.49	0	
B*15:02:01G	1	0.49	0	
B*07:10	1	0.49	0	
B*27:14	1	0.49	0	
B*15:191	1	0.49	0	
B*15:29:01G	1	0.49	0	
B*14:01:01G	1	0.49	0	
B*39:24:01G	1	0.49	0	
B*41:01:01G	1	0.49	0	
B*55:02:01G	1	0.49	0	
C*07:01:01G	43	21.18	7	
C*07:02:01G	43	21.18	4	
C*06:02:01G	44	21.67	3	
C*04:01:01G	42	20.69	3	
C*12:03:01G	41	20.20	4	High predicted SARS-CoV-2 binding capacity [15]
C*02:02:02G	24	11.82	1	
C*01:02:01G	22	10.84	0	Low predicted SARS-CoV-2 binding capacity [15] **We have identified a severe course of ARI ^1^**
C*03:04:01G	17	8.37	0	
C*07:04:01G	15	7.39	0	
C*03:03:01G	15	7.39	0	
C*17:01:01G	11	5.42	0	
C*12:02:01G	11	5.42	0	
C*05:01:01G	10	4.93	2	
C*03:02:01G	10	4.93	0	
C*15:02:01G	9	4.43	0	
C*14:02:01G	5	2.46	0	
C*08:02:01G	4	1.97	0	
C*08:01:01G	4	1.97	0	
C*15:05:01G	3	1.48	0	
C*08:03:01G	2	0.99	0	
C*16:02:01G	2	0.99	0	
C*03:54	1	0.49	0	
C*04:03:01G	1	0.49	0	
C*16:01:01G	1	0.49	0	

^1^ According to the odds ratio (OR), see also Table 5.

**Table 4 viruses-17-01495-t004:** Distribution of HLA-DRB*, HLA-DQA1*, HLA-DQB1*, and HLA-DPB1* alleles frequencies identified in patients with ARI symptoms (number and percentage of patients with the allele, number of homozygotes).

Alleles	Number of Patients	Frequencies, %	Homozygotes	Comments
DRB1*15:01:01G	40	19.70	6	
DRB1*01:01:01G	41	20.20	4	May be a predictor of the development of severe forms of influenza [11]
DRB1*07:01:01G	41	20.20	2	
DRB1*03:01:01G	30	14.78	2	
DRB1*13:01:01G	29	14.29	3	
DRB1*11:01:01G	22	10.84	0	
DRB1*16:01:01G	20	9.85	0	
DRB1*13:02:01G	17	8.37	0	
DRB1*11:04:01G	17	8.37	0	
DRB1*15:02:01G	17	8.37	0	
DRB1*13:03:01G	15	7.39	0	
DRB1*08:01:01G	13	6.40	0	
DRB1*04:01:01G	12	5.91	0	
DRB1*09:01:02G	11	5.42	0	
DRB1*04:04:01G	10	4.93	0	
DRB1*12:01:01G	10	4.93	0	
DRB1*14:01:01G	9	4.43	0	
DRB1*04:02:01G	7	3.45	0	
DRB1*10:01:01G	4	1.97	0	
DRB1*04:03:01G	3	1.48	0	
DRB1*08:03:02G	2	0.99	0	
DRB1*04:08:01G	2	0.99	0	
DRB1*04:07:01G	2	0.99	0	
DRB1*04:05:01G	1	0.49	0	
DRB1*08:02:01G	1	0.49	0	
DRB1*15:02:02G	1	0.49	0	
DRB1*01:02:01G	1	0.49	0	
DRB1*04:06:01G	1	0.49	0	
DRB1*11:03:01G	1	0.49	0	
DRB1*12:02:01G	1	0.49	0	
DRB1*14:04:01G	1	0.49	0	
DRB1*04:101	1	0.49	0	
DRB1*01:03:01G	1	0.49	0	
DRB1*04:10:01G	1	0.49	0	
DRB1*16:02:01G	1	0.49	0	
DRB4*01:01:01G	79	38.92	9	
DRB3*02:02:01G	72	35.47	9	
DRB3*01:01:02G	54	26.60	5	
DRB5*01:01:01G	40	19.70	6	
DRB5*02:02:01G	20	9.85	0	
DRB3*03:01:01G	17	8.37	0	
DRB5*01:02:01G	17	8.37	0	
DRB4*01:02	3	1.48	0	
DRB5*01:10N	1	0.49	0	
DRB3*02:01:01G	1	0.49	0	
DRB4*01:03:03	1	0.49	0	
DQA1*01:02:01G	71	34.98	13	**We have identified the course of ARI with hypoxia ^1^**
DQA1*01:01:01G	55	27.09	7	
DQA1*03:01:01G	50	24.63	4	
DQA1*01:03:01G	46	22.66	4	
DQA1*02:01:01G	41	20.20	2	
DQA1*04:01:01G	13	6.40	0	
DQA1*06:01:01G	2	0.99	0	
DQB1*03:01:01G	69	33.99	4	
DQB1*02:01:01G	59	29.06	10	
DQB1*05:01:01G	46	22.66	7	
DQB1*06:02:01G	38	18.72	5	
DQB1*06:03:01G	29	14.29	4	
DQB1*05:02:01G	23	11.33	0	
DQB1*03:02:01G	22	10.84	2	Associated with moderate or mild MERS-CoV disease [14]
DQB1*03:03:02G	18	8.87	1	
DQB1*06:01:01G	17	8.37	1	
DQB1*04:02:01G	16	7.88	1	
DQB1*06:04:01G	11	5.42	0	
DQB1*05:03:01G	10	4.93	3	
DQB1*06:09:01G	6	2.96	0	
DQB1*03:04:01G	1	0.49	0	
DQB1*03:12	1	0.49	0	
DQB1*03:05:01G	1	0.49	0	
DQB1*05:04:01G	1	0.49	0	
DPB1*04:01:01G	126	62.07	33	
DPB1*02:01:02G	58	28.57	3	
DPB1*04:02:01G	55	27.09	5	**We have identified the course of ARI without hypoxia ^1^**
DPB1*03:01:01G	41	20.20	3	
DPB1*01:01:01G	14	6.90	0	
DPB1*23:01:01G	10	4.93	0	
DPB1*17:01:01G	9	4.43	0	
DPB1*05:01:01G	8	3.94	0	
DPB1*14:01:01G	7	3.45	0	
DPB1*06:01:01G	6	2.96	0	
DPB1*13:01:01G	5	2.46	0	
DPB1*09:01:01G	5	2.46	0	
DPB1*10:01:01G	4	1.97	0	
DPB1*19:01:01G	3	1.48	0	
DPB1*15:01:01G	3	1.48	0	
DPB1*11:01:01G	2	0.99	0	
DPB1*224:01	1	0.49	0	
DPB1*02:02:01G	1	0.49	0	
DPB1*34:01:01G	1	0.49	0	
DPB1*257:01	1	0.49	0	
DPB1*26:01:02G	1	0.49	0	
DPB1*16:01:01G	1	0.49	0	

^1^ According to the odds ratio (OR), see also Table 5.

**Table 5 viruses-17-01495-t005:** Identified alleles associated with different courses of ARI.

Group	Age	PCR	Alleles	Condition ^1^	OR ^2^	CI^2^	Conclusion
Children and adolescents with ARI	0–16	+/−	A*11:01:01G	S/M	5.037	1.717–14.776	severe course of ARI
			A*03:01:01G	Hyp	0.358	0.178–0.72	course without hypoxia
			B*51:01:01G	S/M	4.457	1.355–14.663	severe course of ARI
			C*01:02:01G	S/M	4.743	1.538–14.629	severe course of ARI
			DPB1*04:02:01G	Hyp	0.462	0.244–0.876	course without hypoxia
			DQA1*01:02:01G	Hyp	1.811	1.003–3.268	course with hypoxia
Children with ARI	0–9	+/−	A*11:01:01G	S/M	4.444	1.501–13.159	severe course of ARI
			A*03:01:01G	Hyp	0.301	0.133–0.678	course without hypoxia
			C*01:02:01G	S/M	3.925	1.184–13.006	severe course of ARI
Children with ARVI	0–9	+	A*11:01:01G	S/M	5.654	1.631–19.600	severe course of ARVI
			A*03:01:01G	Hyp	0.317	0.110–0.914	course without hypoxia

^1^ S/M—severe course/moderate course of infection, Hyp—hypoxia/no hypoxia. ^2^
*p* < 0.05.

## Data Availability

The data presented in this study are available on request from the corresponding author due to ethical reasons.

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
