# Peer review of "Different HLA Alleles Frequencies and Their Association with Clinical Phenotypes of Acute Respiratory Infections in Children"

_viruses, 2025, doi:10.3390/v17111495_

Round 1
Reviewer 1 Report (Previous Reviewer 4)
Comments and Suggestions for Authors
Accept in present form
Author Response
Dear Reviewer! We thank you very much for taking the time to review this manuscript and for positive decision.
Reviewer 2 Report (New Reviewer)
Comments and Suggestions for Authors
- L. 124-127: Influenza virus is not included in the common respiratory viruses. To be added.
- L. 136-139: There is no meaning in recording nationalities since numbers are so little and, furthermore, most of the children did not indicate their origins. Nationalities should be recorded in a better designed research, in which the groups of participants would be equal.
- L. 166-167: Why are adolescents not studied as a separate group? Even if this is due to their small number, it should be mentioned and commented.
- L. 173: P-values are not presented at all in the present paper.
- Table 1, Main diagnoses: Seven out of the 18 diagnoses (almost 40%) are not complete, and do not conclude to the responsible viral pathogen. What is the concept of reporting only the rest of the viral pathogens? They could be equally omitted.
- Table 1, Concomitant diagnoses: All kinds of co-morbidities affecting internal systems could be considered as relevant to respiratory conditions, but not myopia!
- Table 2, Legend: Influenza viruses are again omitted. To be added.
- L. 208: Are influenza viruses not detected or are they just omitted by mistake? To be checked.
- L 229-241: Is there any statistical significance in these correlations?
- L. 233-234: Dengue is not a respiratory infection. If this information has to be used, it should be commented in the Discussion.
- Table 4, “Associated with…MERS-CoV disease”: Unlike all the other coronaviruses, MERS-CoV has not been mentioned in Materials and Methods as an investigation target. To be added in Materials and Methods.
- L. 282: Mild cases are still ARI/ARVI.
- L.286: It both impossible and useless to find individuals who have never had any viral infection, since favorable immune profiles do not exclude mild and asymptomatic infections. To be corrected. Notably, all the hospitalized cases can be considered as severe, and “healthy volunteers” could have been selected according to their reported medical history.
- L. 288: All the presently known respiratory viruses do have testing systems. To be corrected.
- L. 310-311: Are these common alleles associated to severity of viral infections?
The use of English should improve.
Author Response
Dear Reviewer,
We thank you very much for taking the time to review the manuscript: Different HLA alleles frequencies and their association with clinical phenotypes of acute respiratory infections in children
Please find the detailed responses below. We appreciate all comments and tried to improve our manuscript according to them.
Comments and Suggestions for Authors:
- 124-127: Influenza virus is not included in the common respiratory viruses. To be added.
Response: Thank you for pointing this out, Enterovirus also was missed. Enterovirus, Influenza A and Influenza B were added to the list.
- 136-139: There is no meaning in recording nationalities since numbers are so little and, furthermore, most of the children did not indicate their origins. Nationalities should be recorded in a better designed research, in which the groups of participants would be equal.
Response: We are thankful for this important suggestion; however, the specifics of the region do not allow the use of nationality in research. We mention ethnicity in the paper, but we consider that the use of ethnicity in this study is unethical and it may mislead researchers. HLA allele and haplotype frequencies in Russia exhibit significant diversity. Russia is home to over 190 ethnic groups and mixed marriages are frequent, which leads to the fact that the definition of ethnicity is simply a matter of self-identification of a person. So we can in no way claim that a person who has indicated his ethnicity actually belongs to this ethnicity.
The genetic heterogeneity of Russians and Siberian population is confirmed by scientific works:
Grahovac, B., Sukernik, R. I., O'hUigin, C., Zaleska-Rutczynska, Z., Blagitko, N., Raldugina, O., Kosutic, T., Satta, Y., Figueroa, F., Takahata, N., & Klein, J. (1998). Polymorphism of the HLA class II loci in Siberian populations. Human genetics, 102(1), 27–43. https://doi.org/10.1007/s004390050650
Kuzmich E.V., Pavlova I.E., Bubnova L.N. Genetic distances between Russians from different country regions and other populations of Russia. Medical Immunology (Russia). 2025;27(3):519-530. (In Russ.) https://doi.org/10.15789/1563-0625-GDB-2886
Khamaganova E.G., Leonov E.A., Abdrakhimova A.R., Khizhinskiy S.P., Gaponova T.V., Savchenko V.G. HLA diversity in the Russian population assessed by next generation sequencing. Medical Immunology (Russia). 2021;23(3):509-522. (In Russ.) https://doi.org/10.15789/1563-0625-HDI-2182
- 166-167: Why are adolescents not studied as a separate group? Even if this is due to their small number, it should be mentioned and commented.
Response: We are thankful for this comment. The adolescents group was small and it was not possible to obtain statistical reliability from such a number. We added this information into the text.
- 173: P-values are not presented at all in the present paper.
Response: The P-value is the abstract line 20 and in the line 265 – in the footer of the Table 5. Thanks to you we also added this information in the main text.
- Table 1, Main diagnoses: Seven out of the 18 diagnoses (almost 40%) are not complete, and do not conclude to the responsible viral pathogen. What is the concept of reporting only the rest of the viral pathogens? They could be equally omitted.
Response: Thank you for this comment. A viral pathogen was identified not in all the cases; the diagnosis of viral infection in this case was based on the clinical picture. In some cases, the patient received multiple diagnoses: with virus specified and additional unspecified sometimes. In cases of co-infection the diagnosis is also unspecified.
- Table 1, Concomitant diagnoses: All kinds of co-morbidities affecting internal systems could be considered as relevant to respiratory conditions, but not myopia!
Response: Thank you for bringing this oversight to our attention. This line was removed from the text.
- Table 2, Legend: Influenza viruses are again omitted. To be added.
Response: Thank you for pointing this out, H3N2 also was missed. Enterovirus also was missed again, but in this case the footer is for abbreviations explanations. Enterovirus, Influenza A and Influenza B are written without abbreviations, so they are not in the footer. Information that H3N2 is Influenza A was added.
- 208: Are influenza viruses not detected or are they just omitted by mistake? To be checked.
Response: Thank you for this comment. In this line we speak about “The most frequently detected viruses with low saturation…” In our work there were 40 cases of hRSv and only 9 cases of influenza, and not all of them were with low saturation. To avoid misunderstanding we corrected the sentence: In our work the most frequently detected viruses with low saturation…
- L 229-241: Is there any statistical significance in these correlations?
Response: Thank you for this comment. We identified alleles that are known to have some correlations from the literature (the references to works are given in square brackets).
- 233-234: Dengue is not a respiratory infection. If this information has to be used, it should be commented in the Discussion.
Response: Thank you for pointing this out, the line was removed from the text.
- Table 4, “Associated with…MERS-CoV disease”: Unlike all the other coronaviruses, MERS-CoV has not been mentioned in Materials and Methods as an investigation target. To be added in Materials and Methods.
Response: Thank you for this comment. MERS-CoV was not an investigation target in this work. In this table for some of alleles we show which correlations with respiratory infections are known from the literature (the references to works are given in square brackets) and which were found in our work (marked with bold and the footer added).
- 282: Mild cases are still ARI/ARVI.
Response: Thank you for pointing this out, corrected to ARI.
- 286: It both impossible and useless to find individuals who have never had any viral infection, since favorable immune profiles do not exclude mild and asymptomatic infections. To be corrected. Notably, all the hospitalized cases can be considered as severe, and “healthy volunteers” could have been selected according to their reported medical history.
Response: We are thankful for this important suggestion, the line was corrected.
Not all the hospitalized cases can be considered as severe. For children, mild cases are usually treated on an outpatient basis, but in the case of moderate course, the decision on hospitalization is made by parents. It is also usually recommended to hospitalize young children, since the increase in symptoms at this age occurs very quickly and inexperienced parents may miss a sharp deterioration in the condition.
- 288: All the presently known respiratory viruses do have testing systems. To be corrected.
Response: We are thankful for this important suggestion, the line was removed. But as we know, new viruses and new variants of viruses are constantly emerging, as we have seen with SARS-CoV-2, for which there were no test systems at the beginning of the pandemic.
- 310-311: Are these common alleles associated to severity of viral infections?
Response: This line shows that the sample characteristics are typical for the region. As far as we know, these alleles are not associated with severity of viral infections.
Thank you again for your attention to our manuscript.

Round 2
Reviewer 2 Report (New Reviewer)
Comments and Suggestions for Authors
Most of the comments were responded; however, the rest of them should be also responded, as follows:
1. Table 2, Legend: “H3N2 – Influenza A H3N2” is wrong. It should be “Influenza A and B viruses”. To be corrected.
2. L 232-244: Comments about statistics should be added regarding the reported findings.
3. L. 242 and Table 4: Unlike all the other coronaviruses, MERS-CoV has not been mentioned in Materials and Methods as an investigation target. To be added in Materials and Methods.
4. L. 310-315: Are these common alleles associated to severity of respiratory infections? The paper studies this association and, therefore, relevant comments should be done.
Author Response
Dear Reviewer,
We thank you very much for taking the time to review the manuscript: Different HLA alleles frequencies and their association with clinical phenotypes of acute respiratory infections in children
Comments and Suggestions for Authors
- Table 2, Legend: “H3N2 – Influenza A H3N2” is wrong. It should be “Influenza A and B viruses”. To be corrected.
Response: In this footer we explain the table abbreviations. H3N2 is a subtype of Influenza A virus not Influenza B.
- L 232-244: Comments about statistics should be added regarding the reported findings.
Response: Thank you for bringing this oversight to our attention. For more clear presentation we added this text: «In the comments column of Tables 3 and 4, we have noted both associations identified in our work (bold font) and associations known from the literature (source references in square brackets)». For alleles identified in our work we also summarized the information about associations known from literature in lines 232-244. So we do not provide statistical data for literature information, only for our own calculations. The text « we found variants for which associations with various courses of infectious diseases are known from the literature» was added to clarify this situation.
- L. 242 and Table 4: Unlike all the other coronaviruses, MERS-CoV has not been mentioned in Materials and Methods as an investigation target. To be added in Materials and Methods.
Response: Thank you for this comment. MERS-CoV was not an investigation target in this work. The alleles which correlate with MERS-CoV are known from the literature (the references to works are given in square brackets).
- L. 310-315: Are these common alleles associated to severity of respiratory infections? The paper studies this association and, therefore, relevant comments should be done.
Response: We are thankful for this important suggestion, we added the text: « Among them, we identified allele A*03:01:01 and show that it is associated with the course of ARVI without hypoxia. The other common alleles from this list show no associations with the course of respiratory diseases».
Thank you again for your attention to our manuscript.
This manuscript is a resubmission of an earlier submission. The following is a list of the peer review reports and author responses from that submission.
Round 1
Reviewer 1 Report
Comments and Suggestions for Authors
In the article, Palyanova et al. investigated the possible association of HLA alleles with the severity of the clinical phenotype of acute respiratory infections in children. The authors performed HLA genotyping in 203 pediatric patients randomly selected from 432 individuals admitted for treatment in Novosibirsk hospitals with signs of acute respiratory infections and concluded that the presence of some allelic variants (in particular, A*11:01:01:01G, A*03:01:01:01G, DQA1*01:02:01G, etc.) is associated with the development of some complications and increased risk of severe course of the disease. The results obtained by the authors for HLA class I are in line with previously published studies, while no data supporting the authors' conclusions for HLA class II are presented in the literature to date. Although the manuscript contains some valuable data, it has a number of serious deficiencies that need to be corrected to make it publishable; these issues are listed below.
First of all, the design of the experiment is vague. It is unclear why the article on the correlation between HLA alleles and severity of respiratory diseases includes clinical data from 432 patients, although the authors did not determine the HLA alleles in all of them, and subsequently randomly selected only 203 (and why so, by the way?) samples for NGS from this sample. Moreover, to talk about the phenotype of ‘acute respiratory viral infections’, it is necessary to deal with PCR-confirmed cases of viral infections, the diagnosis of which by patient’s condition is subjective and unreliable. Here, the authors speculate about the possible association of some HLA alleles and ARVI phenotype, operating on unconfirmed diagnoses (lines 84-87), although in fact respiratory pathology could be caused not by viruses but by bacterial infection/allergies/autoimmune diseases, etc., which is confirmed by 24 types of diagnoses made to patients (lines 120-124). Incidentally, it is unclear why it was not possible to track patients by name or number assigned on first admission to the hospital to further exclude them from NGS sampling after PCR screening results were available, creating a verified ‘infected by viruses’ dataset. Thus, replacements should be made in the manuscript like ‘acute respiratory infections’ to avoid emphasizing of the relationship of HLA alleles to the phenotype of viral infections specifically, since it is not guaranteed that the pathologies of the participants had a viral etiology.
The Abstract and Introduction sections look incomplete and should be supplemented with concluding statements and a justification of the novelty of the research.
Lines 29-33 should be referenced.
Line 45: As the Introduction section deals with different HLA genotypes, information on the classification of HLA classes I and II and their involvement in antiviral/antibacterial immune responses should be provided here.
The term ‘carriage’ (lines 15, 92) is incorrect and should be replaced.
Lines 70-71: why was a sample from a 26 year old donor used in a children study? It seems redundant.
Line 77: patient demographics and histories collected should be given here or in the Results.
The Materials and Methods section would be better divided into subchapters.
Line 81: are you sure the viral DNA was analyzed? Most of the viruses tested are RNA-containing. And why were exactly these 12 viruses screened?
Lines 82-84, 161 and 229 seem to be wrongly constructed and so should be rephrased.
Line 85: The PCR screening procedure should be described (method, instrument, days post symptom onset, etc.).
Line 87: You should describe how the genetic material was isolated.
Line 100: on the contrary, it is necessary to correlate the allele frequencies obtained with ethnic characteristics.
Line 109: The criteria for severe/mild course of ARVI should be explained. Didn't the severe course of the disease include hypoxia? It is not clear, why the hypoxia symptom was considered separately? This question also arises when considering the data in Table 5.
Line 112: It is unclear exactly how OR was calculated.
In the description of the statistical treatment of the results, there is no indication of what criteria were used to test for normality of the data distribution and for intergroup comparisons.
In Table 1, the diagnosis codes should be deciphered and the % of patients should be given, not just the number of participants.
Lines 135-140 contain an almost verbatim paraphrase of Table 2 and are therefore redundant.
Table 2: why was the H3N2 subtype screened separately for influenza?
Saturation, hypoxia, dyspnea, sore throat, headache and intestinal symptoms of patients should also be added to Table 2, as it is difficult to understand the text from lines 149-175. This would also allow the relationship between clinical features and the type of viral infection to be reflected.
It is unclear why lines 198-211 repeatedly list the most frequently identified HLA class II alleles if they are already presented in Table 4. And in contrast, the ‘not only mass ones’ mentioned in lines 213-217 and 219-222 should be added to Table 4.
There is no speculation in the Discussion about the results of analyzing the clinical data of patients and the viruses detected in samples, as well as no mention of study limitations and future prospects.
Line 250: since the phenotype/performance of the patient is not a basis for diagnosing viral infection, it should be explained on the basis of which additional non-PCR tests the diagnosis was made.
Lines 257-262 and 286-289 are irrelevant to this study and should be removed.
Lines 269-270: actually, the paper did not link HLA alleles to the severity of either acute respiratory viral infection itself or severe COVID-19.
The list of references needs to be expanded and updated as it is rather modest and contains mainly articles published before 2020.
Author Response
Comments and Suggestions for Authors
- In the article, Palyanova et al. investigated the possible association of HLA alleles with the severity of the clinical phenotype of acute respiratory infections in children. The authors performed HLA genotyping in 203 pediatric patients randomly selected from 432 individuals admitted for treatment in Novosibirsk hospitals with signs of acute respiratory infections and concluded that the presence of some allelic variants (in particular, A*11:01:01:01G, A*03:01:01:01G, DQA1*01:02:01G, etc.) is associated with the development of some complications and increased risk of severe course of the disease. The results obtained by the authors for HLA class I are in line with previously published studies, while no data supporting the authors' conclusions for HLA class II are presented in the literature to date. Although the manuscript contains some valuable data, it has a number of serious deficiencies that need to be corrected to make it publishable; these issues are listed below.
Response: Thank you very much for taking the time to review this manuscript. Please find the detailed responses below. We appreciate all comments and tried to improve our manuscript according to them. The article has been significantly revised: data not related to patients who did not participate in genotyping have been excluded, and the list of references has been expanded.
- First of all, the design of the experiment is vague. It is unclear why the article on the correlation between HLA alleles and severity of respiratory diseases includes clinical data from 432 patients, although the authors did not determine the HLA alleles in all of them, and subsequently randomly selected only 203 (and why so, by the way?) samples for NGS from this sample.
Response: Thank you for bringing this oversight to our attention. We have revised the article and removed clinical data of 432 patients. Only data from patients included in the final study were retained. During the initial examination in the emergency department, blood tests were taken, anamneses were collected and preliminary diagnoses were made. 432 patients received a preliminary diagnosis of ARVI, since a respiratory tract infection was detected, but a blood test did not confirm a bacterial infection or allergy. Patients for genetic analysis were randomly selected among these 432 patients.
Moreover, to talk about the phenotype of ‘acute respiratory viral infections’, it is necessary to deal with PCR-confirmed cases of viral infections, the diagnosis of which by patient’s condition is subjective and unreliable. Here, the authors speculate about the possible association of some HLA alleles and ARVI phenotype, operating on unconfirmed diagnoses (lines 84-87), although in fact respiratory pathology could be caused not by viruses but by bacterial infection/allergies/autoimmune diseases, etc., which is confirmed by 24 types of diagnoses made to patients (lines 120-124). Incidentally, it is unclear why it was not possible to track patients by name or number assigned on first admission to the hospital to further exclude them from NGS sampling after PCR screening results were available, creating a verified ‘infected by viruses’ dataset.
Response: Thank you for this valuable suggestion. We agree that the presence of a positive PCR test for viral infections is indisputable evidence of a viral infection. However, the absence of a positive PCR test cannot be proof of the absence of a viral infection, since we only test a limited number of viruses, and mutations can help the virus not only avoid attack by the immune system, but also prevent detection by tests. For example, our panel of viruses does not include Epstein-Barr virus, also influenza and SARS-CoV-2 mutate rapidly. In this case, a complete blood count is an additional source of information about the viral or bacterial nature of the disease, which helps reduce the unnecessary use of antibiotics. Thus, we were confident that all patients included in the genetic study had a viral infection. However, to be completely sure, after your request we contacted the medical institutions and clarified the diagnoses made at discharge, the information received was also added to the article. Despite the fact that patients received 24 different diagnoses, almost all of them related to viral infections, the diagnosis decoding was added to the table. Thanks to the clarified diagnoses, we excluded from the study 7 patients with identified concomitant HIV infection, mycoplasma infection and inborn lung disease, and recalculated the statistics again. We also created a verified ‘children under 10 years old infected by viruses’ dataset.
Thus, replacements should be made in the manuscript like ‘acute respiratory infections’ to avoid emphasizing of the relationship of HLA alleles to the phenotype of viral infections specifically, since it is not guaranteed that the pathologies of the participants had a viral etiology.
Response: We are thankful for this important suggestion, we replaced the term ARVI with ‘acute respiratory infections’ for groups of patients with negative PCR.
- The Abstract and Introduction sections look incomplete and should be supplemented with concluding statements and a justification of the novelty of the research.
Response: Thank you for pointing this out. The abstract is only 200 words long, which does not allow for notes on novelty and conclusion, but we have added this information to the introduction and discussion.
- Lines 29-33 should be referenced.
Response: Thank you for bringing this oversight to our attention. The references were added.
- Line 45: As the Introduction section deals with different HLA genotypes, information on the classification of HLA classes I and II and their involvement in antiviral/antibacterial immune responses should be provided here.
Response: We are thankful for this suggestion, the information was added.
- The term ‘carriage’ (lines 15, 92) is incorrect and should be replaced.
Response: Thank you for pointing this out, the term was replaced.
- Lines 70-71: why was a sample from a 26 year old donor used in a children study? It seems redundant.
Response: Thank you for bringing this oversight to our attention, this person was excluded from the sample.
- Line 77: patient demographics and histories collected should be given here or in the Results.
Response: We are thankful for this important suggestion. The demographics and histories were added. We added division into groups by age and PCR, and added the list of symptoms to the table with detected viruses.
- The Materials and Methods section would be better divided into subchapters.
Response: Thank you for this comment. We changed the subchapters division.
- Line 81: are you sure the viral DNA was analyzed? Most of the viruses tested are RNA-containing. And why were exactly these 12 viruses screened?
Response: We are thankful for this important comment. The term was corrected to DNA/RNA.
- Lines 82-84, 161 and 229 seem to be wrongly constructed and so should be rephrased.
Response: Thank you for bringing this oversight to our attention, the sentences were rephrased.
- Line 85: The PCR screening procedure should be described (method, instrument, days post symptom onset, etc.).
Response: We are thankful for this important suggestion. The PCR screening procedure description was added
- Line 87: You should describe how the genetic material was isolated.
Response: We are thankful for this important suggestion. The description of genetic material isolation was added. Genetic material from patients was obtained according to generally accepted protocols from buccal and pharyngeal swabs
- Line 100: on the contrary, it is necessary to correlate the allele frequencies obtained with ethnic characteristics.
Response: Thank you for this comment .HLA allele and haplotype frequencies in Russia exhibit significant diversity. Russia is home to over 190 ethnic groups and mixed marriages are frequent, which leads to the fact that the definition of ethnicity is simply a matter of self-identification of a person. We mention ethnicity in the paper, but we consider that the use of ethnicity in this study is unethical and it may mislead researchers. Let me give you an example: in a situation where, in Russia, the child's mother belongs to one ethnicity and the father to another, the child is most often assigned the ethnicity of Russian, even if both parents are not Russian. When such a “Russian” has own children, they can receive either the ethnicity of one of the older relatives or the ethnicity of the second parent, but they can also remain Russian, which is what most often happens. This situation occurs in many families over many generations, so we can in no way claim that a person who has indicated his ethnicity has at least 50% belonging to this ethnicity.
The genetic heterogeneity of Russians is confirmed by scientific works:
Kuzmich E.V., Pavlova I.E., Bubnova L.N. Genetic distances between Russians from different country regions and other populations of Russia. Medical Immunology (Russia). 2025;27(3):519-530. (In Russ.) https://doi.org/10.15789/1563-0625-GDB-2886
Khamaganova E.G., Leonov E.A., Abdrakhimova A.R., Khizhinskiy S.P., Gaponova T.V., Savchenko V.G. HLA diversity in the Russian population assessed by next generation sequencing. Medical Immunology (Russia). 2021;23(3):509-522. (In Russ.) https://doi.org/10.15789/1563-0625-HDI-2182
We used allele frequencies from [Khamaganova E.G., et. al.] as a reference sample for estimation of HLA allele frequencies in Russian population.
- Line 109: The criteria for severe/mild course of ARVI should be explained. Didn't the severe course of the disease include hypoxia? It is not clear, why the hypoxia symptom was considered separately? This question also arises when considering the data in Table 5.
Response: We are thankful for this important comment. The severity of the disease is determined according to clinical recommendations, which are sufficiently extensive that it does not allow to include them in the article. Although hypoxia is one of the criteria for disease severity, patients in severe condition do not always suffer from hypoxia, and not all patients with hypoxia are classified as severe. We became interested in hypoxia separately from the severe condition, since hypoxia is often the cause of the patient's worsening condition and leads to the development of further complications. That is why the hypoxia symptom was considered separately.
- Line 112: It is unclear exactly how OR was calculated.
Response: Odds ratios were calculated using four-field tables. The odds ratio is the value of a fraction whose numerator contains the odds of a certain event for the first group, and whose denominator contains the odds of the same event for the second group. To assess the significance of the odds ratio, the boundaries of the 95% confidence interval are calculated. If the confidence interval does not include 1, i.e. both boundary values are either above or below 1, a conclusion is made about the statistical significance of the identified relationship between the factor and the outcome at a significance level of p<0.05. The calculations were performed with Medstatistic online calculator.
- In the description of the statistical treatment of the results, there is no indication of what criteria were used to test for normality of the data distribution and for intergroup comparisons.
Response: We used statistical methods for categorical data for which normality testing could not be performed.
- In Table 1, the diagnosis codes should be deciphered and the % of patients should be given, not just the number of participants.
Response: We are thankful for this important suggestion. The Table was reworked: diagnosis codes were deciphered, the % of patients was added. Also we added concomitant diagnoses of patients.
- Lines 135-140 contain an almost verbatim paraphrase of Table 2 and are therefore redundant.
Response: Thank you for bringing this oversight to our attention, the lines were shortened.
- Table 2: why was the H3N2 subtype screened separately for influenza?
Response: Our test systems detect unspecified influenza A and influenza B, then, if there is sufficient material, the virus is typed. Thus, influenza A was detected in 6 unspecified cases and 11 cases of H3N2, H1N1 was not detected. Influenza B unspecified - 3 cases.
- Saturation, hypoxia, dyspnea, sore throat, headache and intestinal symptoms of patients should also be added to Table 2, as it is difficult to understand the text from lines 149-175. This would also allow the relationship between clinical features and the type of viral infection to be reflected.
Response: We are thankful for this important suggestion, the table was expanded.
- It is unclear why lines 198-211 repeatedly list the most frequently identified HLA class II alleles if they are already presented in Table 4. And in contrast, the ‘not only mass ones’ mentioned in lines 213-217 and 219-222 should be added to Table 4.
Response: Thank you for bringing this oversight to our attention, the lines 198-211 were removed. We also added ‘not only mass alleles’ into the table.
- There is no speculation in the Discussion about the results of analyzing the clinical data of patients and the viruses detected in samples, as well as no mention of study limitations and future prospects.
Response: Thank you for bringing this oversight to our attention, the discussion was reworked.
- Line 250: since the phenotype/performance of the patient is not a basis for diagnosing viral infection, it should be explained on the basis of which additional non-PCR tests the diagnosis was made.
Response: We are thankful for this important suggestion. The explanation was added to methods.
- Lines 257-262 and 286-289 are irrelevant to this study and should be removed.
Response: Thank you for bringing this oversight to our attention, the lines 257-262 were removed. The lines 286-289 refer to alleles that in our work have been shown to be related to respiratory diseases. We suggest that information about their association with other diseases is important.
- Lines 269-270: actually, the paper did not link HLA alleles to the severity of either acute respiratory viral infection itself or severe COVID-19.
Response: The lines 269-270 are about the allele A*11:01:01:01. According to literature is linked to severe COVID-19 and in our work we show the association of this allele not only with COVID-19, but with respiratory viral infections.
- The list of references needs to be expanded and updated as it is rather modest and contains mainly articles published before 2020.
Response: Thank you for this valuable suggestion. Work on HLA is quite rare and labor-intensive, so in some cases there are only a few studies related to a specific allele or disease.

Reviewer 2 Report
Comments and Suggestions for Authors
The manuscript by Palyanova et al. studies potential genetic susceptibility factors for pediatric respiratory infections, specifically named as acute viral respiratory infections (ARVI). The authors collected during 9 months a cohort of 432 patients in medical institutions of Novosibirsk with clinical symptoms of acute respiratory disease, diagnosed across 24 different ICD codes. Together with available clinical data, to identify possible etiological agents, PCR testing for 12 common viral pathogens was conducted. Genetic analysis focused on HLA loci using Illumina sequencing for 203 individuals. Ethnic background information was also collected for a part of cohort. The study compares HLA allele frequencies between patient's groups with severe versus mild disease and between different levels of hypoxia.
Major Criticism:
The manuscript lacks a clearly defined research goal and suffers from weak structuring, making it difficult to understand the study’s design and scientific aim. As an observational study, it is largely descriptive and limited by poor design and insufficient statistical power. Attempting to detect genetic associations across hundreds of HLA variants in a clinically heterogeneous cohort without consideration of appropriate population controls, treatment variables, patient age is unlikely to yield reliable results.
No statistical models incorporating relevant covariates are presented. Instead, the analysis is limited to crude OR calculations without adjustments for multiple comparisons.
Study Design Issues:
The manuscript fails to articulate a primary hypothesis or research question.
Inclusion and exclusion criteria are not clearly stated.
Criteria used to define disease severity are not described. Were severity assessments consistent across different age groups (e.g., infants vs. adolescents)? Were individuals with immunodeficiencies excluded? Was it performed according to standardized protocol in different medical institutions?
Pooling patients with 24 different diagnoses and multiple ethnic backgrounds, without justification, undermines the scientific validity of the study.
The authors do not discuss the minimum sample size required to test multiple hypotheses related to genetic associations.
Ethnic variation in HLA distribution can result in spurious associations if not appropriately controlled, but this issue was considered in the study as irrelevant.
Hypoxia Analysis:
The comparison of HLA alleles based on levels of hypoxia is undermined by a lack of methodological detail. It is unclear how hypoxia was defined or measured, whether the method was validated and standardized between clinics, at what point in the disease course measurements were taken. Hypoxia can fluctuate significantly with time and treatment, and is more likely influenced by respiratory pathophysiology rather than by HLA-related immunological mechanisms.
Diagnostic and Etiological Concerns:
The rationale for grouping different etiologies under the broad category of ARVI is unclear. This may reflect local diagnostic conventions but diverges from internationally accepted clinical practice. No justification is provided for this approach.
The use of pathogen detection data is also problematic. Detection of viral or bacterial nucleic acids does not necessarily confirm causation, particularly for ubiquitous pathogens and may result in confusing findings.
Hospitalisation by itself indicates disease severity, but the authors do not clarify why the patients with the "moderate" disease course were hospitalized: was it a primer diagnosis or other conditions that coincided with ARVI.
Potential confounding factors such as age, socioeconomic status, and healthcare access are not discussed and not used in statistical evaluation.
In summary, due to multiple flaws in study design and lack of statistical robustness, the findings are not reliable and the conclusions are not convincing.
Minor Comments:
HLA gene nomenclature is inconsistent. Standardized nomenclature should be used throughout, following international guidelines.
Comments on the Quality of English LanguageThe manuscript requires extensive language editing by a native English speaker or professional editor.
Author Response
Comments and Suggestions for Authors
- The manuscript by Palyanova et al. studies potential genetic susceptibility factors for pediatric respiratory infections, specifically named as acute viral respiratory infections (ARVI). The authors collected during 9 months a cohort of 432 patients in medical institutions of Novosibirsk with clinical symptoms of acute respiratory disease, diagnosed across 24 different ICD codes. Together with available clinical data, to identify possible etiological agents, PCR testing for 12 common viral pathogens was conducted. Genetic analysis focused on HLA loci using Illumina sequencing for 203 individuals. Ethnic background information was also collected for a part of cohort. The study compares HLA allele frequencies between patient's groups with severe versus mild disease and between different levels of hypoxia.
Response: Thank you very much for taking the time to review this manuscript. Please find the detailed responses below. We appreciate all comments and tried to improve our manuscript according to them. The article has been significantly revised, data not related to patients who did not participate in genotyping have been excluded.
Major Criticism:
- The manuscript lacks a clearly defined research goal and suffers from weak structuring, making it difficult to understand the study’s design and scientific aim. As an observational study, it is largely descriptive and limited by poor design and insufficient statistical power. Attempting to detect genetic associations across hundreds of HLA variants in a clinically heterogeneous cohort without consideration of appropriate population controls, treatment variables, patient age is unlikely to yield reliable results.
Response: Thank you for bringing this oversight to our attention. The manuscript was significantly revised. The research goal and the scientific aim have been revised and presented more clearly. The aim of this study was to determine the allelic composition and allele frequencies in children hospitalized with signs of acute respiratory infection, as well as to identify the relationship between the disease phenotype and the patient's genotype. HLA genotyping and identification of patients with alleles associated with severe respiratory diseases or with a risk of hypoxic conditions will help the doctor decide on treatment tactics and prescribe treatment for such patients without waiting for the condition to worsen. Population studies in Siberia are rare and limited, so this work is a pilot for Siberia. The statistical power of the study was increased thanks to your recommendations: we divided patients into groups depending on age and the presence of a positive PCR.
- No statistical models incorporating relevant covariates are presented. Instead, the analysis is limited to crude OR calculations without adjustments for multiple comparisons.
Response: We are thankful for this comment. This is a pilot study and the odds ratio is a sufficient method at this stage.
Study Design Issues:
- The manuscript fails to articulate a primary hypothesis or research question.
Response: We are thankful for this important suggestion. Our research interest was to find alleles that influence the course of the disease. We expanded the introduction to disclose it.
- Inclusion and exclusion criteria are not clearly stated.
Response: Thank you for bringing this oversight to our attention. Inclusion and exclusion criteria were added.
- Criteria used to define disease severity are not described. Were severity assessments consistent across different age groups (e.g., infants vs. adolescents)? Were individuals with immunodeficiencies excluded? Was it performed according to standardized protocol in different medical institutions?
Response: We are thankful for this important suggestion. The criteria we added. The severity of the disease was determined based on clinical guidelines, according to standardized protocol for different medical institutions which included clinical and laboratory parameters differentiated depending on the patient's age. The individual with HIV was excluded.
- Pooling patients with 24 different diagnoses and multiple ethnic backgrounds, without justification, undermines the scientific validity of the study.
Response: We are thankful for this important suggestion. Although there are many diagnoses, they are all associated with respiratory infections. The only difference is in the severity, localization and expression of symptoms and the type of virus that causes the disease. The Table of diagnoses was reworked: diagnosis codes were deciphered, the % of patients was added. Also we singled out separately concomitant diagnoses of patients.
We mention ethnicity in the paper, but we consider that the use of ethnicity in this study is unethical and it may mislead researchers. HLA allele and haplotype frequencies in Russia exhibit significant diversity. Russia is home to over 190 ethnic groups and mixed marriages are frequent, which leads to the fact that the definition of ethnicity is simply a matter of self-identification of a person. So we can in no way claim that a person who has indicated his ethnicity actually belongs to this ethnicity.
- The authors do not discuss the minimum sample size required to test multiple hypotheses related to genetic associations.
Response: The required sample size for testing multiple genetic association hypotheses is influenced by the number of SNPs being tested, the minor allele frequency, the effect size of the genetic variants, the study design, and the desired statistical power and significance level. Because this is a pilot study and we do not know all the characteristics of the population, it is not possible to reliably calculate the required sample size.
- Ethnic variation in HLA distribution can result in spurious associations if not appropriately controlled, but this issue was considered in the study as irrelevant.
Response: HLA allele and haplotype frequencies in Russia exhibit significant diversity. The genetic heterogeneity of Russians is confirmed by scientific works:
Kuzmich E.V., Pavlova I.E., Bubnova L.N. Genetic distances between Russians from different country regions and other populations of Russia. Medical Immunology (Russia). 2025;27(3):519-530. (In Russ.) https://doi.org/10.15789/1563-0625-GDB-2886
Grahovac, B., Sukernik, R. I., O'hUigin, C., Zaleska-Rutczynska, Z., Blagitko, N., Raldugina, O., Kosutic, T., Satta, Y., Figueroa, F., Takahata, N., & Klein, J. (1998). Polymorphism of the HLA class II loci in Siberian populations. Human genetics, 102(1), 27–43. https://doi.org/10.1007/s004390050650
Khamaganova E.G., Leonov E.A., Abdrakhimova A.R., Khizhinskiy S.P., Gaponova T.V., Savchenko V.G. HLA diversity in the Russian population assessed by next generation sequencing. Medical Immunology (Russia). 2021;23(3):509-522. (In Russ.) https://doi.org/10.15789/1563-0625-HDI-2182
We used allele frequencies from [Khamaganova E.G., et. al.] as a reference sample for estimation of HLA allele frequencies in Russian population.
Hypoxia Analysis:
- The comparison of HLA alleles based on levels of hypoxia is undermined by a lack of methodological detail. It is unclear how hypoxia was defined or measured, whether the method was validated and standardized between clinics, at what point in the disease course measurements were taken. Hypoxia can fluctuate significantly with time and treatment, and is more likely influenced by respiratory pathophysiology rather than by HLA-related immunological mechanisms.
Response: We are thankful for this important suggestion. Hypoxia was established upon admission to the health care facility, before the start of treatment. The criteria for hypoxia were added to the text. The level of hypoxia is indeed closely related to the level of respiratory damage, while the damage itself is closely related to the immune response (sufficient, insufficient, excessive).
Diagnostic and Etiological Concerns:
- The rationale for grouping different etiologies under the broad category of ARVI is unclear. This may reflect local diagnostic conventions but diverges from internationally accepted clinical practice. No justification is provided for this approach.
Response: We are thankful for this important comment. All the patients in the study received a preliminary diagnosis of ARVI, since a respiratory tract infection was detected, but a blood test did not confirm a bacterial infection or allergy. Internationally accepted clinical practice is to use PCR test to confirm viral infections. We agree that the presence of a positive PCR test for viral infections is indisputable evidence of a viral infection. However, the absence of a positive PCR test cannot be proof of the absence of a viral infection, since we only test a limited number of viruses, and mutations can help the virus not only avoid attack by the immune system, but also prevent detection by tests. For example, our panel of viruses does not include Epstein-Barr virus, also influenza and SARS-CoV-2 mutate rapidly. In this case, a complete blood count is an additional source of information about the viral or bacterial nature of the disease, which helps reduce the unnecessary use of antibiotics. Thus, we were confident that all patients included in the genetic study had a viral infection. However, to be completely sure, we contacted the medical institutions and clarified the diagnoses made at discharge, the information received was also added to the article. Thanks to the clarified diagnoses, we excluded from the study 7 patients with identified concomitant HIV infection, mycoplasma infection and inborn lung disease, and recalculated the statistics again. We also created a verified ‘children under 10 years old infected by viruses’ dataset.
- The use of pathogen detection data is also problematic. Detection of viral or bacterial nucleic acids does not necessarily confirm causation, particularly for ubiquitous pathogens and may result in confusing findings.
Response: We are thankful for this important suggestion. The mentions of viral infection were removed from irrelevant places in text. Therefore, the diagnosis is made not only on the basis of PCR, but also taking into account the anamnesis and blood tests.
- Hospitalisation by itself indicates disease severity, but the authors do not clarify why the patients with the "moderate" disease course were hospitalized: was it a primer diagnosis or other conditions that coincided with ARVI.
Response: For children hospitalization of severe cases always occurs, mild cases are usually treated on an outpatient basis, but in the case of moderate course, the decision on hospitalization is made by the parents. It is also usually recommended to hospitalize young children, since the increase in symptoms at this age occurs very quickly and inexperienced parents may miss a sharp deterioration in the condition. In our study ARVI was a primer diagnosis.
- Potential confounding factors such as age, socioeconomic status, and healthcare access are not discussed and not used in statistical evaluation.
Response: In Russia, all children receive medical care free of charge, regardless of socioeconomic status. We tried to take into account the influence of age and added this information to the article.
- In summary, due to multiple flaws in study design and lack of statistical robustness, the findings are not reliable and the conclusions are not convincing.
Response: Thank you again for your attention to our manuscript. We have significantly revised the article, introduced division into groups by age and PCR results into the statistical analysis, which improved the design of the study and made the results and conclusions more reliable.
Minor Comments:
- HLA gene nomenclature is inconsistent. Standardized nomenclature should be used throughout, following international guidelines.
Response: Thank you for pointing this out. The standardized HLA gene nomenclature was added.
Comments on the Quality of English Language
- The manuscript requires extensive language editing by a native English speaker or professional editor.
Response: The language editing was performed.

Reviewer 3 Report
Comments and Suggestions for Authors
Review Manuscript ID: viruses-3726269, titled "Frequencies and association of different HLA alleles with clinical phenotypes of acute respiratory viral infections in children" by Natalia V. Palyanova et al.
Below are the comments to the authors:
Major comments
- The authors used CBC (Complete Blood Count) for infection risk stratification to enroll subjects with ARI. This method, not well defined by the authors, entails a serious conditioning on the scientific validity of the manuscript. In fact, if we adopt an enrollment methodology based on the subject's immune response it will then be difficult to demonstrate that the HLA alleles are distributed equally in the sample.
- The negative samples for the 12 viruses subjected to RT-PCR were 55, the authors explain with what criterion they enrolled them as positive for ARI with viral etiopathogenesis. it is known that j20.9 and j06.9 are typically caused by a viral infection, although bacterial infections can also be a cause.
- The authors explain what it refers to when reporting "seasonal influenza"
Minor comments
- Line 14: .....and performed genetic analysis for them... I hope it was actually for them!
- Line 21 and 22: Please define the meaning of ARVI
- The abstract is not structured as recommended, perhaps the conclusions are missing
- to indicate gender, male and female are usually used and not boy or girl, please correct the sentence
- In table 1, there were 203 positive classified subjects enrolled in total. compared to the 195 declared, of which 55 had unspecified etiopathogenesis.
Reviewer 4 Report
Comments and Suggestions for Authors
In the present form this manuscript cannot be fairly evaluated. In the present form it has to be rejected.
There are two major problems:
- The copy I have received is cluttered with correction marks.
- Presentation of data - many, poorly organized Tables - make it impossible to evaluate the data and the derived conclusions.
Please consult with an expert in style and presentation of data
Round 2
Reviewer 1 Report
Comments and Suggestions for Authors
The authors have substantially improved the manuscript in accordance with the reviewers' comments. In particular, they have detailed the Introduction and Discussion, removed redundant clinical data, corrected inaccuracies in the methodology description and improved the presentation of the results in the table form. In my opinion, the article may be published in its current form, but I recommend moving large Tables 3 and 4 to the Supplementary, leaving only their brief description in the main text, in order to shorten the manuscript and make the presentation of the main results as clear as possible.
Reviewer 2 Report
Comments and Suggestions for Authors
Although several technical issues have been addressed appropriately, the overall aim of the study and the key aspects of the study design have only been marginally improved in the revised manuscript. My major concerns regarding population admixture and phenotype definitions remain, and therefore the overall conclusions of the study are not convincingly supported. I appreciate the authors’ efforts in conducting this investigation; however, I would recommend a more focused approach and a more careful consideration of ethnicity definitions. Studying a multiethnic population in a genetic context is challenging and possibly requires comprehensive genetic profiling beyond the HLA locus.